# The Connectivity of the Resting Brain in Primary Open-Angle Glaucoma: A Systematic Review

**DOI:** 10.3390/biomedicines13061402

**Published:** 2025-06-07

**Authors:** Nikola Velkov, Sevdalina Kandilarova, Drozdstoy Stoyanov

**Affiliations:** 1Faculty of Medicine, Medical University of Plovdiv, 4000 Plovdiv, Bulgaria; 2Department of Psychiatry and Medical Psychology, Research Institute and SRIPD-MUP, Medical University of Plovdiv, 4000 Plovdiv, Bulgaria; sevdalina.kandilarova@mu-plovdiv.bg (S.K.); drozdstoy.stoyanov@mu-plovdiv.bg (D.S.)

**Keywords:** primary open-angle glaucoma, neuroimaging, resting-state functional magnetic resonance imaging

## Abstract

**Background/Objectives**: Worldwide, glaucomas are the leading cause of irreversible blindness in adults. On the ocular level, they are fairly well understood; however, the functional and structural changes that occur in the brain have become a subject of great interest lately, mostly owing to improved accessibility and effectiveness of functional magnetic resonance imaging (fMRI). This, coupled with the non-invasive nature of the methodology, has contributed to an ever-growing body of research published on the topic. In this systematic review, we gather, systematize, and compare the results and methodologies reported in the literature, as pertaining to resting-state fMRI brain changes in primary open-angle glaucoma (POAG). **Methods**: A systematic search in PubMed, Scopus, and Web of Science was carried out, resulting in a total of 290 records identified, with 67 assessed for eligibility and 24 selected for inclusion. **Results**: The main findings include worse functional parameters in the early visual centers in POAG across all methodologies, reduced functional connectivity between V1 and other parts of the visual cortex, functional aberrations in higher levels of the visual system, predominantly in the ventral stream and in extravisual networks, among others. Moreover, the majority of these changes are shown to be correlated with ophthalmological measurements. **Conclusions**: Although studies on this matter tend to suffer from a limited sample size and a lack of methodological standardization, we nevertheless manage to present common results and conclusions regarding the effects of POAG on brain function.

## 1. Introduction

Glaucoma is a term referring to a broad group of progressive multifactorial optic neuropathies, characterized by an excavated appearance of the optic disc, loss of retinal ganglion cells, and their axons and corresponding visual loss [1]. It is the leading cause of irreversible blindness globally with circa 60 million people affected in 2014, with projections for 2040 surpassing 111 million individuals worldwide [2,3].

The main (although not mandatory) risk factor for glaucoma is constantly elevated intra-ocular pressure (IOP) ≥ 21 mmHg. Generally, glaucomas are classified into open and closed angle regarding the presence or lack of visible obstruction of the trabecular meshwork in the anterior chamber, which is the start of the main outflow path for aqueous humor and therefore has a main role in IOP regulation. They are further etiologically classified into primary and secondary [1]. Overall, primary open-angle glaucoma (POAG) is by far the most common variant, accounting for about 3/4 of cases worldwide [4].

The core of this pathology lies in the apoptotic degeneration of the retinal ganglial cells (RGCs), which leads to gradual loss of vision, from peripheral to central, and eventually blindness. This results from an aberration of axonal transport of neurotrophic factors within the axons of the RGCs, which pass through a mechanically compressed, posteriorly displaced, and thinned Lamina Cribrosa, following long-term exposure to increased IOP. Clinically, these changes are often assessed through direct ophthalmoscopy—“cupping” of the disc, increased cup-to-disk ratio (CDR), or through optical coherence tomography—reduced retinal nerve fiber layer thickness (RNFLT) [1,5].

Scientific interest lately has focused on the structural and functional changes in the visual pathway beyond the optic nerve as well as in other cortical areas and networks. In one of the first controlled studies with such scope, Chaturvedi et al., 1993 [6], discovered that, histologically, the magnocellular layers in glaucoma patients were significantly atrophied as compared to controls. Later, multiple MRI studies showed that this anatomical lateral geniculate nucleus (LGN) atrophy can be effectively studied in living subjects through neuroimaging [7,8]. Since then, various methods have been developed, most broadly being separated into structural—detailing the volume and composition of relevant brain areas—and into functional, which tracks changes in blood flow/oxygenation and, therefore, activation in different parts of the brain following a specific visual stimulus (task-based) or the functional interactions between different regions without stimuli (resting-state) [9].

At present, two methods have been widely adopted for capturing resting-state functional magnetic resonance (RS-fMRI) data—Blood-Oxygenated-Level–Dependent (BOLD) and arterial spin labeling (ASL)—with each approach being based on different physiological processes and providing data with distinct qualities. BOLD signal relies on the difference in magnetic susceptibility between oxygenated and deoxygenated hemoglobin, based upon which blood flow in different brain parts may be qualitatively assessed with a relatively high degree of temporal resolution [10]. ASL uses magnetically labeled arterial blood water protons as an endogenous tracer, which allows for a direct quantitative measurement of localized cerebral blood flow (CBF), at the cost of lower temporal resolution [11].

Common measures for quantifying and interpreting BOLD data include functional connectivity (FC), amplitude of low-frequency fluctuations (ALFFs), and regional homogeneity (ReHo). FC is currently one of the most widely used fMRI methodologies, applicable both in RS and task-based studies. It examines the strength and organization of connectivity on the global or network level, based on the temporal correlation of BOLD signal fluctuations between separate regions. The simplest approach to this end is seed-based FC, which is driven by a preconceived hypothesis—a specific voxel or region of interest’s time series is correlated to all other voxels in a subject’s brain. Whole-brain analysis is a data-driven approach, during which a connectivity matrix is set up, allowing examination of activation correlations between any two regions without a predetermined ROI. This offers great exploratory potential at the cost of increased computational demand [12].

Independent component analysis (ICA) is another data-driven approach, which decomposes fMRI data into temporally correlated statistically independent components, which may then be used as basis of further analyses [13]. Dynamic functional connectivity (DFC) involves dividing the BOLD time series into time windows, calculating the FC for each and comparing them, and enabling the quantification of variability/stability across time on an ROI, network, or whole-brain level [14].

On the other hand, ALFF is a summation of amplitudes of each voxel’s signal frequency spectrum within the low-frequency range (0.01 to 0.08/0.1 Hz) and reflects the amplitude of spontaneous low-frequency BOLD signal fluctuations, i.e., the baseline brain function [15,16]. ReHo measures the local synchronization of BOLD signal fluctuations across the time series of a voxel and its neighbors using Kendall’s coefficient of concordance. Simply put, it assesses local coherence of activity, highlighting separate functionally specialized areas [17]. In graph theory, the parcellated regions are assigned as nodes, while the connections between them are seen as edges. From then on, various network characteristics are calculated, reflecting the intricate behaviors of complex brain systems [18].

We believe that a deeper understanding of the functional changes in subcortical visual pathways, the visual network, and their relationships to extravisual regions and networks resulting from glaucoma will contribute to the field of neuroscience and in the future could translate to the development of more holistic and reliable prevention and treatment regimens. In this paper we would like to focus specifically on RS-fMRI analyses performed on the most common type of glaucoma—the POAG—by exploring and comparing their methodologies and results. To our knowledge, in the last 15 years, only three reviews regarding this topic have been published. Garaci et al., 2015 [19], produced a narrative review curating results from a broad spectrum of neuroimaging modalities in glaucoma. In 2018, Nuzzi et al. [20] published a systematic review cataloging results from structural, functional, and metabolic brain imaging for various types of glaucomas. Lately, Sujanthan et al. [21] produced a narrative review, which examines results exclusively in the visual system regarding multiple optic neuropathies. As there has been significant technological advancement and a large number of new publications on the topic in the last years, we surmised that a new focused and systematic paper, which also discusses methodology, is warranted.

## 2. Materials and Methods

### 2.1. Registration

We registered this systematic review in the Open Science Framework (OSF) under doi.org/10.17605/OSF.IO/U5S3Y. We then completed the 27-item PRISMA checklist.

### 2.2. Search Strategy

We performed a systematic search in three databases (PubMed, Scopus, and Web of Science) for articles published until 6 December 2024 with the terms “glaucoma”, “functional MRI”, “resting state”, and their derivative terms (more detailed information about our search strategy can be found in Appendix A). We also explored and included suitable references cited in articles we examined.

### 2.3. Selection

We included papers meeting the following criteria: affected population was adult humans, diagnosed with high-tension primary open-angle glaucoma, age- and gender-matched controls were present, original research published in English language, and presence of RS-fMRI data. Exclusion criteria were lack of high-tension POAG diagnosis, even with any other type of glaucoma or an unspecified such present in patient population, lack of matched controls, papers with a patient group <5 individuals, animal-only studies, lack of RS-fMRI data, reviews and other kinds of publications without original data, non-English-language literature. The two researchers independently screened the articles, preparing a selection list based on aforementioned criteria. Mismatches between the selections were discussed and corrected.

### 2.4. Extraction and Analyses

The full texts of the final selection of articles were carefully examined to extract the following information: authors, year of publication, main aim of the study, description of RS-fMRI methodology utilized, type of comparison groups and their sizes, main results from the neuroimaging and clinical correlation analysis. The data extracted has been used to produce tables and figures for better understanding and appropriate presentation of the results of the current review.

## 3. Results

Our search strategy with the above-mentioned search terms in the three databases resulted in the identification of a total of 290 records, 147 of which were duplicates. Additionally, 1 study was identified from citation searching. Of the 143 records screened, 76 were excluded because they were not relevant to our review aim. All 67 remaining records were successfully retrieved and assessed for eligibility, and 44 of these were excluded because they did not correspond to our predetermined criteria—paper type not being original research, not containing a population with POAG diagnosis, not containing RS-fMRI data, adult human subjects, or a control population. Finally, 24 records fitting all our criteria were selected for inclusion in the systematic review. A detailed overview of the search procedure is given in the PRISMA flow chart (Figure 1).

In order to present the results of the included studies in a more orderly fashion, we segregated them into four subsections with corresponding tables based on the main methodology utilized. Section 3.1 and Table 1 include studies that rely on FC and its variants, and Section 3.2 and Section 3.3 and Table 2 and Table 3 review the studies using ALFF and ReHo, respectively. In Section 3.4 and Table 4, ASL-CBF papers are summarized, and Section 3.5 and Table 5 show the ones employing graph theoretical analysis techniques.

**Table 1 biomedicines-13-01402-t001:** Classical functional connectivity studies.

Author, Year	Stated Aim	Subject	Method	Main Finding in POAG	Clinical Correlation
Dai et al., 2013 [22]	Analyze FC changes in visual system in POAG	22 POAG 22 HC	Seed-based, controlled for atrophy through Voxel-Based Morphometry	↓ positive FC b/w BA17 and R ITG, L MOG, L postCG, L preCG; b/w BA18/19 and vermis, R MTG, R STG; ↑ negative FC b/w BA17 and anterior cerebellar lobe; ↓ negative FC b/w BA17 and R MiFG, R middle cerebellar peduncle, left cerebellum, BA18/19, and R Ins	
Frezzoti et al., 2014 [23]	Assess white matter tracts integrity, gray matter volume changes, FC network changes, relationship to visual impairment	13 POAG 12 HC	pICA, visual selection, voxelwise analysis	↓ FC in exstrastriate VN (R LG), WMN (L SFG, R SMG, R LOC), DAN (b LOC, L preCG, L postCG); ↑ FC in VN (b LOC, L Fusiform), medial EN (R SFG, paracingulate, R AC)	↓ MD—↓ FC in (b Precun, R Cun, R Calc, R MFG, R SPL)
Frezzotti et al., 2016 [24]	Assess if diffuse brain changes shown in advanced POAG can be detected since the early stage using multimodal MRI	57 POAG 14 early 13 interm. 30 adv. 29 HC	pICA, voxelwise analysis	↓ FC in VN (R Fusiform; R ITG; R LOC), WMN (R PCC; L ITG; R AG) ↑ FC in DMN (L LOC), SCN (L Putamen)	↓ FC in LOC—↑ PSD
Zhou et al., 2016 [25]	Explore changes in interhemispheric FC through VMHC	25 POAG 15 HC	VMHC (FC between symmetric interhemispheric voxels)	↓ VMHC in Calc, Cun, Precun; ↑ VMHC in Ins, SMG, frontal gyrus, LG	↓ VMHC in Precuneus— ↑ CDR
Wang, J. et al., 2016 [26]	Explore the alterations of FC and subnetwork connectivity of the VN and DMN	25 POAG 25 HC	ICA for VN and DMN, followed by FC and FNC	↓ FC in V1 ↓ FNC V1-V2; V1-DMN ↑ FNC V2-DMN	FNC ↑ V1-DMN—↑ MD on the left
Giorgio et al., 2017 [27]	Relate presence or absence of raised IOP to neurodegenerative findings in normal-tension glaucoma and POAG	17 POAG 10 mild 2 mod. 5 severe 17 NTG 29 HC	Multimodal, including ICA, followed by voxelwise FC	↑ FC in medial frontal ECN, VAN ↑ FNC b/w V2 and LN	—
Wang, Q et al., 2018a [28]	Investigate if abnormal VMHC is accompanied by anatomic connectivity changes and relate it with ophthalmic parameters	16 POAG 6 early 4 interm. 6 adv. 19 HC	VMHC, homotopic Diffusion Tensor Imaging, correlation	↓ zVMHC in BA17 (V1), BA18 (V2), BA19 (V3,4,5)	VMHC ↑ BA17, BA18, BA19—↑ RNFLT mean
Wang, Y. et al., 2020 [29]	Assess alterations in resting-state visual networks in patients with POAG and investigate the effect of elevated IOP	36 POAG 20 HC	ICA with spatial correlation analysis for 3 visual networks, intranetwork FC	↓ FC b/w L Calc—lateral network ↓ FC b/w b LG—Medial network ↓ FC b/w b LG—occipital network	↓ FC b/w L Calc—lateral network—↑ IOP
Wang, B et al., 2021 [30]	Evaluate the effect of elevated IOP on FC of the VN	36 POAG 20 HC	Voxelwise FC analysis of 1 ROI	↓ FC b/w BA17—R SFG; BA17—R Precun	↓ FC (BA17—R SFG)—↑ IOP
Yang et al., 2024 [31]	Analyze brain functional abnormalities through Dynamic FC—functional stability	70 POAG 45 HC	Dynamic FC—functional stability (Kendall’s coefficient)	↓ Stability in early visual centers (V2, V3, V4), dorsal stream, ventral stream ↑ Stability in b IPL and R inferior frontal cortex	↓ Stability in L early visual centers and dorsal stream—↓ MD

FC—Functional Connectivity; VMHC—Voxel-Mirrored Homotopic Connectivity; ICA—Indepent Component Analysis; ROI—Region Of Interest; POAG—patients; HC—Heallthy Controls; ↓—decrease; ↑—increase, b/w — between; RNFLT—Retinal Nerve Fiber Layer Thickness; MD—Mean Deviation of Visual Field; CDR—Cup-to-Disk Ratio; IOP—Intra-Ocular Pressure; BA —Brodmann’s area; ITG—inferior temporal gyrus; MOG—middle occipital gyrus; PostCG—postcentral gyrus; PreCG—precentral gyrus; MTG—middle temporal gyrus; STG—superior temporal gyrus; MiFG—middle frontal gyrus; Ins—insula; LG—lingual gyrus; SFG—superior frontal gyrus; SMG—supramarginal gyrus; LOC—lateral occipital cortex; PCC—posterior cingulate cortex; AG—angular gyrus; Calc—calcarine cortex; Cun—cuneus; and Precun—Precuneus. Networks: VN—visual; WMN—working memory; DAN—dorsal attention; EN—executive; ECN—extracortical; VAN—ventral attention; and VMHC—voxel-mirrored homotopic connectivity.

### 3.1. Classical Functional Connectivity Techniques

Ten of the studies selected for this review used some form of FC as their primary analysis method (Table 1). The patient sample varied widely from 13 (Frezotti et al., 2014 [23]) to 70 (Yang 2024 [31]). Most studies reported decreased FC in the striate (Brodmann area BA17/V1 [22,24,25,26,28,30]) and extrastriate (BA18, BA19/V2, V3, V4 [22]) visual cortices to both visual networks (VNs) and extravisual regions and voxels. Of note are also the reduced connectivity in parts of the ventral stream, especially in the R inferior temporal gyrus (ITG) [22,24,31], and to a lesser extent in the dorsal stream [31]. Multiple articles also reported FC increase in extravisual networks, most notably in the default mode network (DMN) [24,26]. Most studies also correlated their findings with ophthalmological measurements—visual field loss (Mean Deviation—MD), IOP, RNFLT, CDR or glaucoma stage—with reduced FC in striate and extrastriate cortices being associated with clinical markers, characteristic for a more advanced glaucomatous process or the risk of such.

In 2013, Dai et al. [22] published the first study examining RS-fMRI aberrations in POAG. They conducted seed-based analyses with BA17 (V1), BA18 (V2), BA19 (V3, V4, V5/middle temporal visual area (MT)), and BA7 (intraparietal cortex, part of the dorsal stream) as ROIs, while controlling for brain atrophy. Although no differences in the FC measures between these four ROI were discovered, ROI-to-voxel analysis reported many significant results, which were classified by the authors according to whether FC had been positive or negative and whether it had increased, decreased, appeared, or disappeared. Decreased positive FC was noted between BA17 and R ITG. BA17-positive FC to L postcentral gyrus (postCG) and L precentral gyrus (preCG) was also decreased as well as BA18/BA19 to R middle temporal gyrus (MTG) and R superior temporal gyrus (STG).

Frezzoti et al. in 2014 [23] and in 2016 [24] used a multimodal MRI approach to look into structural and functional network changes in POAG. They used ICA and voxelwise analysis, followed by a clinical correlation to the degree of visual field defect. The 2014 study, which had a rather small sample size, consisting of 13 patients and 12 controls, reported decreased FC in regions such as the R lingual gyrus (LG), part of the visual network (VN), the L postCG, L preCG, R lateral occipital cortex (LOC), and part of the dorsal attention network (DAN). Decreased FC was also noted in L superior frontal gyrus (SFG), R supramarginal gyrus (SMG), and R LOC, all part of the working memory network (WMN).

The next study from 2016 used a much larger sample size, including 57 patients at various stages of disease progression and 29 healthy controls [24]. In all stages of POAG, in the VN, the FC in the R inferior LOC, R ITG, and R Fusiform gyrus decreased. Similar to the 2014 paper, decreased FC in the WMN—R posterior cingulate cortex (PCC), L ITG, R angular gyrus (AG)—was noted. Some differences in the early-stage glaucoma as compared to all-stage are described—namely the lack of reduced FC in R Fusiform (VN), R AG (WMN), lack of increased FC in L LOC (DMN); however, FC was reduced in bilateral (B) MTG (DMN), B frontal poles, and L paracingulate (both WMN).

Zhou et al., 2016 [25], and Wang, Q et al., 2018a [28], used voxel-mirrored homotopic connectivity (VMHC) to explore interhemispheric FC in POAG. The first study discovered a decreased degree of VMHC between the calcarine cortex (Calc), cuneus (Cun) (corresponding to V1 and V2), and Precuneus (Precun), which is responsible for a multitude of highly integrated tasks. VMHC was increased in insula (Ins), SMG, frontal gyrus (FG), and LG. The second article came up with similar results—reduced VMHC in BA17 (V1), BA18 (V2), and BA19 (V3,4,5). The authors propose that besides the small sample sizes, the younger population, a new MRI machine, and using gray matter as a covariate in the newer article contribute to the discrepancy in results.

Wang, J et al., 2016b [26], performed a group ICA, isolating components within the VN and DMN, whose FC was then investigated. Only one component in the occipital pole (corresponding to V1) showed a significantly decreased FC, which is consistent with Frezotti 2014 and 2016 results. The authors also conducted a functional network connectivity analysis between the ICs, noting a decreased value between the ICs for V1 and V2 and between V1 and various DMN components.

The report by Giorgio et al., 2017 [27], focuses primarily on changes in normal-tension glaucoma, but it also includes a standard high-tension POAG cohort. It once again used ICA, followed by voxelwise FC analysis. Compared to HC, POAG displayed higher FC within the executive control network (ECN) and the ventral attention network (VAN), as well as between V2 and the limbic network. Interestingly, when compared to the normal-tension group, POAG had higher FC within the VN and the ECN.

Wang, Y. et al., 2020 [29], uniquely used three ICA-based visual subnetworks to measure FC: medial VN, which includes BA17 and BA18 (V1 and V2); lateral VN, BA19 (V3, V4, V5/MT), lateral occipital, and superior occipital gyri; and occipital VN associated with higher level processing. According to their analysis, the network FC between L Calc and lateral VN, between LG and medial VN, and between LG and occipital VN was reduced.

In 2021, Wang, B. et al. [30] employed an ROI-based FC RS-fMRI approach to examine the functional connectivity of BA17 to the rest of the brain within the same patient and control population as Wang, Y et al., 2020 [29]. They reported a decrease in FC between BA17 and R Precun (DMN); BA17 and R SFG (WMN).

Among the selected studies, only one used DFC, and it was published in 2024 by Yang et al. [31]. They discovered decreased stability in early visual centers (V2, V3, V4) as well as in various parts of the dorsal and ventral visual streams. Increased stability was noted in bilateral inferior parietal lobe (IPL) and in R inferior frontal cortex.

**Table 2 biomedicines-13-01402-t002:** ALFF-based studies.

Author, Year	Stated Aim	Subject	Method	Main Finding in POAG	Clinical Correlation
Liu and Tian, 2014 [32]	To investigate regional spontaneous activity and correlate with disease severity	21 POAG 22 HC	ALFF, correlation with HAP	↓ ALFF in R V1; R Fusiform; R LG; R ITG; L postCG; L preCG; R posterior CL; ↑ ALFF in R MeFG and R SMA	↑ ALFF in R SFG and ↓ ALFF in L occipital; L postCG—↑ HAP
Li et al., 2014 [33]	To analyze altered ALFF	21 POAG 22 HC	ALFF and fALFF, correlation with HAP	↓ ALFF IN R LG, R ITG, L preCG; ↑ ALFF in R MiFG, R SMA; ↓ fALLF in b Cun, R MTG, R PostCG, L PCC, R lymbic lobe; ↑ fALLF in R middle cingulate cortex, L IPL, R MiFG	↑ Spont. activities in L Cun, b MTG ↓ in R SFG—↑ severity (HAP)
Yuan et al., 2018 [34]	To determine the role of the locus coeruleus–norepinephrine system in POAG in patients through ALFF and FC and experimentally in animals	22 POAG 22 HC + animals	ALFF for LC, seed-based FC for LC, clinical correlations	↑ ALFF in LC ↑ FC between LC and parahippocampus ↓ FC between LC and R Ins and R frontal lobe	↑ ALFF in LC—↑ CDR; MD; ↓ RNFLT

ALFF—Amplitude of Low Frequency Fluctuations; fALLF—fractional ALFF; FC—Functional Connectivity; POAG—patients; HC—heallthy controls; HAP— Hodapp-Anderson-Parrish scale; CDR—Cup-to-Disk Ratio; RNFLT—Retinal Nerve Fiber Layer Thickness; ↓—decrease; ↑—increase; LG—lingual gyrus; ITG—inferior temporal gyrus; PostCG—postcentral gyrus; PreCG—precentral gyrus; CL—cerebellar lobe; MeFG—medial frontal gyrus; SMA—superior motor area; MiFG—middle frontal gyrus; Cun—cuneus; MTG—middle temporal gyrus; PCC—posterior cingulate cortex; IPL—intra parietal lobe; and Ins—insula.

### 3.2. ALFF-Based Techniques

We included three papers based on ALFF, which used two patient populations, containing, respectively, 21 [32,33] and 22 [34] POAG subjects. Their findings are summarized in Table 2.

The report by Liu and Tian, 2014 [32], is a conference paper that demonstrated decreased ALFF in R V1, R Fusiform, R LG, R ITG, as well as in L PreCG, L PostCG, and R cerebellar posterior lobe (CPL). ALFF was increased in R superior motor area and in R medial frontal gyrus (MeFG). Later, the same group expanded on their research and produced a full original article (Li et al., 2014) [33]. The results reported there were similar, with ALFF decreasing in R LG, R ITG, and L PreCG and increasing in R MiFG and R SMA. Another metric—fractional ALFF (fALLF) for slow four and five bands, which is less susceptible to physiological artifacts—was also calculated in Li et al., 2014 [33]. fALLF was increased in B Cun, R MTG, R PostCG, L PCC, and R lymbic lobe and decreased in the R middle cingulate cortex, L IPL, and R MiFG.

In 2018, Yuan et al. [34] published a complex multimodal study, which focused exclusively on the role of the locus coeruleus (LC) in POAG and included experimental animal data as well. ALFF in LC in the POAG group increased and was correlated to all clinical markers used, except IOP. FC between LC and parahippocampus increased, and between LC and R Ins and R frontal lobe, it decreased.

**Table 3 biomedicines-13-01402-t003:** ReHo-based studies.

Author, Year	Stated Aim	Subject	Method	Main Finding in POAG	Clinical Correlation
Song et al., 2014 [35]	To investigate spontaneous activity in POAG	39 POAG 41 HC	ReHo, clinical correlation with visual field	↑ ReHo in R anterior cingulate cortex, B MeFG, R anterior CL, R SFG ↓ ReHo in B Calc, R LG, B Precun, B preCG, B postCG, L IPL, L posterior CL	↑ ReHo in SFG, L IPL, L Calc and ↓ in Precun—↑ MD
Wang, Y et al., 2019 [36]	To evaluate the effects of high-IOP on CNS in POAG	36 POAG 20 HC	ReHo, clinical correlation with IOP	↑ ReHo in L CL VIII (posterior), CL IV, CL V (anterior), L Fusiform ↓ ReHo in L MiFG	↑ ReHo in L Fusiform and ↓ in MiFG—↑ IOP

ReHo—Regional Homogeneity; POAG—Patients; HC—Heallthy Controls; MD—Mean Deviation of visual field; ↓—decrease; ↑—increase; MeFG—medial frontal gyrus; CL—cerebellar lobe; SFG—superior frontal gyrus; Calc—calcarine sulcus; LG—lingual gyrus; Precun—Precuneus; PreCG—precentral gyrus; PostCG—postcentral gyrus; IPL—inferior parietal lobule; and MiFG—middle frontal gyrus.

### 3.3. ReHo-Based Techniques

As for ReHo, two studies were included, both containing larger subject groups—39 [35] and 36 [36] patients (Table 3). Firstly, Song et al. [35] discovered increased ReHo in the R anterior cingulate cortex, B MeFG, R SFG, and R anterior cerebellar lobe. Decreased ReHo was observed in bilateral Calc, R LG, bilateral Precuneus, PreCG, PostCG, L IPL, and L posterior CL. Several correlations with MD were observed, most notably between decreased ReHo in the Precuneus and increased MD.

In 2019, Wang, Y et al. [36] found decreased ReHo in L MiFG, another region mentioned by multiple ALFF and FC studies. Increased ReHo was observed in L Fusiform; however, what is most interesting is the increased ReHo in both anterior (lobes IV and V) and posterior (lobe VIII) cerebellar lobes, which contradicts Song et al.’s findings, who reported the ReHo in posterior CL to be reduced.

**Table 4 biomedicines-13-01402-t004:** ASL-CBF-based studies.

Author, Year	Stated Aim	Subject	Method	Main Finding in POAG	Clinical Correlation
Zhang et al., 2015 [37]	To assess the cortical structure and cerebral blood flow changes in POAG	23 POAG (subdivided) 29 HC	ASL-CBF in resting state, then with task, Voxel-Based Morphometry	Only in advanced disease: ↓ CBF in anterior Calc	
Wang, Q et al., 2018b [38]	To investigate the correlations between reduced CBF and changes in the retinas mild-to-moderate POAG through ASL-CBF	15 (mild-to-moderate) POAG 20 HC	ASL-CBF in resting state, comparison for interhemispheric symmetricity, correlation with CDR, RNFLT, GCC	In mild and moderate disease, ↓ zCBF in L V1, L V2, R Ventral posterior area (V3v), L LOC ↑ zCBF symmetricity in V1 and V3v/VP	↓ zCBF in R V3v/VP, R V2—↑ CDR, ↓ GCC, RNFLT
Wang, Q et al., 2021 [39]	To test whether disturbed neurovascular coupling in visual and higher-order cognitive cortices exists in POAG and correlates with disease stage and VF defects	45 POAG 12 early 19 intermed. 4 advanc. 25 HC	ASL-CBF in resting state, ratio with FC strength (CBF/FCS), correlation with stage and MD	↓ CBF/FCS in b LG; b Calc; b Rectal gyri; R STG; R ITG; R IFG ↑ CBF/FCS in R AG; R MiFG ↓ CBF in B LG; B Calc; R PostCG; R IFG; R SMG; B IPL; L SMG, B cerebellum ↑ CBF in B Rectal gyri; B MiFG; R MeFG; R SFG; R Ins	↓ CBF/FCS in b LG—↑ stage/severity, MD defect
Wang, Q et al., 2024 [40]	To investigate CBF-redistributed patterns in visual and higher-order cognitive cortices and its clinical correlations	45 POAG 23 HC	ASL-CBF, CBF connectivity (CBFC)	↓ CBF in b LG, b Calc, R PostCG, R IPL, L cerebellar crus, R CL VI (posterior CL) ↑ CBF in R medial prefrontal gyrus, R MeFG, b MiFG, R SFG; R Ins ↓ negative CBFC in R mPFC—R ITG, R MOG Appeared negative CBFC between PostCG and R Calc, R SOG; R SFG and R ITG; L IPL and R STG Appeared positive CBFC in L MFG—R CPL, R ITG; L MFG—R CPL, R ITG; R MFG—R CPL, R MTG	↓ CBF in b LG, b Cal and ↑ CBF in L cerebellar crus, MeFG—↑ MD

ASL—Arterial Spin Labeling; CBF—Cerebral Blood Flow; POAG- patients; HC—Heallthy Controls; CDR—Cup-to-Disc Ratio; RNFLT—Retinal Nerve Fiber Layer Thickness; GCC—Ganglion Cell Complex thickness; ↓—decrease; ↑—increase; Calc—calcarine sulcus; LOC—lateral occipital cortex; STG—superior temporal gyrus; IFG—inferior frontal gyrus; ITG—inferior temporal gyrus; AG—angular gyrus; MiFG—middle frontal gyrus; PostCG—postcentral gyrus; SMG—supramarginal gyrus; IPL—inferior parietal lobule; MeFG—medial frontal gyrus; SFG—superior frontal gyrus; Ins—insula; MOG—middle occipital gyrus; SOG—superior occipital gyrus; and mPFC—medial prefrontal cortex.

### 3.4. ASL-CBF-Based Techniques

The present review included four studies measuring CBF through ASL (Table 4). One of these also measured BOLD-FC, ratioing the two parameters [39]. Zhang et al., 2015 [37], combined RS and task-based ASL-CBF analyses with voxel-based morphometry. The RS part found no CBF aberrations in subjects with early and intermediate disease; however, in advanced cases, reduced CBF in the anterior part of the calcarine fissure was reported, which was also complemented by reduced gray matter volume in the same region.

The three following studies were all published by the same Beijing-based group. In Wang, Q, 2018b [38], 15 patients with mild-to-moderate POAG were included. Decreased CBF was reported in L V1, L V2, R ventral posterior area (V3v), and L LOC, which fits in with Zhang et al.’s results. Furthermore, increased interhemispheric symmetricity of CBF in V1 and V3v was also noted.

Wang, Q et al., 2021 [39], used a larger patient population in various stages of POAG. CBF was reported lower in bilateral LG, bilateral Calc, R PostCG, R IFG, R SMG, bilateral IPL, and bilateral cerebellum and higher in bilateral RG, bilateral MiFG, R MeFG, R SFG, and R Ins. The methodology used was identical to the previous article. BOLD-FC was also measured, being reduced in R AG, R MiFG, and L cerebellar cruz. CBF was then ratioed with local FC to give clues to regional neurovascular coupling. CBF/FC ratios were reported increased in B LG, B Calc, B Rectal gyri, R STG, R ITG, and L IFG and increased in R AG and R MFG. When limited to early POAG, bilateral LG showed reduced CBF/FCS and only R AG increased.

Recently, Wang, Q et al. [40] used data from the same subject group yielding mostly the same results, despite the slightly different statistical corrections used. As in the 2021 article, CBF in bilateral LG, bilateral Calc, R PostCG, bilateral IPL, and parts of the cerebellum (R CL VI, cerebellar crus) decreased, and in the R MeFG, bilateral MiFG, R SFG, R Ins, it increased. The authors selected 11 regions for ROIs, based on them displaying significant CBF differences between the tested populations. The CBF in these ROIs was then correlated in pairs, producing CBF connectivity (CBFC) maps. Decreased negative CBFC was between R medial prefrontal cortex and R ITG as well as R MOG. In contrast to controls, POAG patients showed negative CBFC between R PostCG and R Calc, R superior occipital gyrus; between R SFG and R ITG; and between L IPL and R STG. Positive CBFC was recorded between L MFG and R CPL, R ITG; between L MFG and R CPL, R ITG; and between R MFG and R CPL, R MTG.

All three studies from the Beijing group also correlated CBF (2018 and 2024) and CBF/FCS (2021) with ophthalmological measures. Generally, worse clinical markers were correlated with reduced neuroimaging markers within visual centers and with increased such in extravisual regions.

**Table 5 biomedicines-13-01402-t005:** Graph theory-based studies.

Author, Year	Stated Aim	Subject	Method	Main Finding in POAG	Clinical Correlation
Wang J. et al., 2016a [41]	To investigate the efficiency of the functional communication change in POAG	25 POAG 25 HC	GTA at 13% sparsity, BC, Deg, Eg, El used as topological properties at global and local levels, correlation to MD and CDR	Global GT metrics—no significant differences ↓ Disruption indices ↓ BC in L IFG, L Fusiform, L hippocampus, and R paracentral lobule ↑ BC in R MFG, L SMA, R Amygdala, R AG, L Thalamus, R Heschl’s gyrus 6 hub regions not in POAG—L IFG, B Fusiform, L Precun, L STG, R MTG 9 hub regions only in POAG—L preCG, R SFG/MiFG, R IFG, R Hyppocampus, R Amygdala, R LG, L MOG, R STG	↓ BC of R Fusiform; ↑ BC of R LG—↓ R MD
Minosse et al., 2019 [42]	To evaluate the potential of functional network disruption indices as biomarkers of disease severity	19 POAG 16 HC	GTA at 10% sparsity, calculation of BC, El, spectral measure of centrality, clinical correlation	Global and local GT metrics—no differences ↓ Disruption indices 2 hub regions not in POAG—R AG, L CL VII 3 hub regions only in POAG—R inferior occipital cortex, R ITG, L CL IX	Positive association between disruption indices and MD, macula ganglion cell layer thickness, RNFLT
Qu. et al., 2020 [43]	To generate a visual atlas based on FC from POAG patients and to prove its applicability on FC and network analysis	36 POAG 20 HC	Parcellation of visual cortex, GTA using a data-driven atlas—Deg, E, SWI, RCI at global and nodal levels as topological properties	↓ Deg, rich club index ↑ Nodal E, small-world index More asymmetric parcellation in visual cortices	—
Demaria et al., 2021 [44]	To determine network integrity in glaucoma and ways in which the VF could affect the hub function of networked brain areas	20 POAG 24 HC	Two scans, fast eigenvector centrality mapping, hub selection (5% highest EC), correlation	Global and local functional networks—no differences Aberrant EC in Ins and MiFG in 1 of 2 scans	Aberrant EC in R LG with ↓ binocular integrated visual field

GTA—Graph Theorhetical Analysis; POAG—patients; HC—Heallthy Controls; MD—Mean Deviation of visual field; CDR—Cup-to-Disk Ration; BC—betweenness centrality; Deg—degree; Eg—global efficiency; El—local efficiency; SWI—small-world index; EC—eigenvector centrality; ↓—decrease; ↑—increase; IFG—Inferior Frontal Gyrus; MiFG—Middle Frontal Gyrus; SMA—Superior Motor Area; AG—Angular Gyrus;Precun—Precuneus; STG—Superior Temporal Gyrus; preCG—Precentral Gyrus; SFG—Superior Frontal Gyrus; LG—Lingual Gyrus; MOG—Middle Occipital Gyrus; STG—Superior Temporal.

### 3.5. Graph Theory-Based Techniques

We included four articles that used some form of graph theory analysis, with patient population varying from 19 [42] to 36 [43] subjects. Given the small number of papers and the significant differences in methodology and aims between studies, it is understandable that the results in this section are not well comparable. We’ve summarised them in Table 5.

Wang, J et al., 2016a [41], was the first study to use graph theory to examine brain changes in glaucoma. Lack of significant differences in global network measurements between POAG and HC was reported. However, a decrease in all disruption indices and aberrations in local network measurements such as betweenness centrality (BC), degree (Deg), local efficiency (El), and global efficiency (Eg) was found. Focusing on BC (the fraction of all shortest paths in the network that pass through a given node), they reported it to be decreased in L IFG, L Fusiform, L hippocampus, and R paracentral lobule and increased in R MFG, L SMA, R Amygdala, R AG, L Thalamus, and R Heschl’s/Transverse temporal gyrus. They also identified six hub regions present in HC but not in POAG—L IFG, B Fusiform, L Precuneus, L STG, R MTG—and nine hub regions present only in POAG—L preCG, R SFG/MiFG, R IFG, R hippocampus, R Amygdala R LG, L MOG, and R STG. Aberrant BC in R Fusiform and R LG was well correlated with increased MD of the right visual field.

Minosse et al., 2019 [42], similarly to Wang, J et al., 2016a [41], reported no differences in global network measurements as well as group-wise differences in subject-wise disruption indices in local metrics. Identification of hub regions, however, came with different results—observed only in HC were R AG, and L CL VII and only in POAG patients, R inferior occipital cortex, R ITG, and L CL IX were observed—none of which correspond to the previous study’s findings.

In a largely methodological study, Qu et al., 2020 [43], attempted to generate a functional brain atlas of the visual cortices in POAG in patients. They then performed GTA on the Taian patient group with the goal of testing their atlas, reporting their results rather concisely. In POAG, compared to HC, they discovered reduced Deg and rich club index and increased nodal efficiency and small-world index in the visual cortices.

Demaria et al., 2021 [44], performed two scans per subject, followed by a whole-brain FC analysis and fast eigenvector centrality (EC) mapping. Hubs were selected based on them having an eigenvalue above the 95th percentile. In only one of the two scans, EC difference between hubs of the two groups reached significance—namely in the insula and MFG (parts of the salience network). Weak correlations were also noted between the EC of R LG and visual field indices.

## 4. Discussion

Overall, our review highlights several cerebral RS functional changes in patients suffering from POAG. Firstly, in the early visual cortices, functional activity measures were significantly and consistently reduced. Secondly, in a multitude of the included papers, functional changes in higher parts of the visual network were reported, with a bias for the ventral or “what” pathway. Thirdly, some non-visual networks, most notably the default mode network, also seem to undergo functional aberrations, resulting from the glaucomatous process. And finally, many of these results were significantly correlated with clinical markers for glaucoma severity, which showcased the relationship between degeneration of visual capabilities and neural function.

### 4.1. The Visual System

The visual system is probably the most complex of all the human sensory apparatuses. After the optical components of the eye focus the light on the retina, the observed image is transduced into neural signal, which passes through the RGCs and LGN to reach the primary visual cortex (V1), where the initial stage of visual processing is carried out. Signal then continues to the secondary visual cortex (V2), where more advanced processing is undertaken as well as feedback modulation of V1 activity. Within these two early regions begins a gradual divergence of informational pathways into two distinct streams with separate functions. The dorsal stream passes through V3, V5/MT, and the intraparietal area, terminating in the motor cortex and is responsible for visuo-spatial awareness and coordination of actions. The ventral stream goes through V4, the temporo-occipital, and inferotemporal lobe in order to reach the prefrontal and parahippocampal cortex, delivering detailed information about form and structure of the visual components [45,46,47].

### 4.2. RS Changes in the Visual Cortex

Many studies have reported functional and structural degenerative changes in the lower parts of the visual pathway in glaucoma, most notably in the optic nerve and the LGN [48]. With this in mind, we expected similar findings regarding the primary visual cortex. Aberrations in the function of V1 were reported in studies with various methodologies—reduced ALFF (spontaneous function) in Li et al., 2014 [33], and its complementary conference paper [32] and reduced ReHo (specialization) in Song et al., 2014 [35]. All four of the included ASL-based articles [37,38,39,40,41] also reported reduced CBF in V1, and three of these (except for Zhang et al., 2015 [37]) in parts of the extrastriate visual cortex (V2, V3, V4, V5/MT). Furthermore, two studies [25,28] found reduced interhemispheric connectivity between symmetrical bilateral regions within the entire early visual cortex. On a similar note, in an ASL-based study, Wang, Q et al., 2018 [38], reports increased interhemispheric symmetricity of CBF in V1 and in V3v.

Furthermore, connectivity between V1 and the extrastriate visual cortex was lower in multiple studies (Dai et al., 2013 [22], Wang, J 2016b [26], Wang, Y 2020 [29], Wang, B 2021 [30]), which indicates a trans-synaptic degeneration of the visual pathway and decreased transmission of visual information between primary and higher visual cortices. Frezotti et al., 2014 [23], also reported decreased network connectivity in the R LG (V2).

### 4.3. RS Changes in VN Components Outside the Visual Cortices

Many studies have shown that functional aberrations within the VN are not limited to the visual cortices and continue to higher cerebral levels. In their more recent paper, Frezzotti et al., 2016 [24], reported decreased network connectivity in the R LOC and the Fusiform gyrus, both being implicated in high-order visual processing, i.e., the recognition of complex objects and faces [49,50].

As for the dorsal stream, there was one study that specifically reported signal changes within it—reduced dynamic stability in Yang et al., 2024 [31]. On the level of the LGN, there is some data that points to POAG being associated with a stronger structural degradation within magnocellular layers, compared to parvocellular [6,51]. The magnocellular layers take mostly wide peripheral visual input, the loss of which is considered the hallmark symptom of glaucoma [1]. As the dorsal stream is responsible for awareness, orientation, coordination of the body, and action guidance within space [45,46], the intuitive presumption is that its input would stem predominantly from the magnocellular pathway and would therefore have significant functional aberrations in POAG patients. Even though there is evidence suggesting magno- and parvocellular input to dorsal stream to be roughly equal in strength [52], we still found the lack of reports of functional changes there surprising.

Within the ventral stream, on the other hand, a multitude of studies using various modalities have reported different functional aberrations, mostly concentrated on the right inferior temporal gyrus. This is proposed to be associated with deficits in visual memory consolidation and memory-related imagery [46,47]. Furthermore, the R ITG is an integrative center for visual, auditorym and tactile information and could relate to impairments in sensory integration. Dai et al., 2013 [22], reported reduced R ITG V1 functional connectivity, while Liu and Tian, 2014 [32] and Li et al., 2014 [33], found reduced ALFF. Utilizing ASL, Wang, Q et al. demonstrated lower CBF/FC (neurovascular coupling) [39] and reduced CBF connectivity [40] in this region. In Minosse et al., 2019 [42], ITG appeared as a network hub in POAG patients but not in controls.

Overall, according to the reports we have included, it may be surmised that in resting state, the ventral stream is more significantly affected by chronic glaucoma, as compared to the dorsal stream. However, the intricacies regarding the relationship between the glaucomatous eye, the changes in the lower parts of the visual system, and the higher-order visual cortical regions remain yet to be further explored.

### 4.4. RS Changes in Extravisual Networks

The effect of the chronic glaucomatous process seems to not be limited to the visual network. Some of the papers included in our review point to a wider functional involvement of several brain networks, including the default mode, salience, working memory, subcortical, ventral and dorsal attention networks, among others. Of these, however, the DMN seems to be most affected. Generally, it is responsible for internally focused thought processes and memory during lack of external stimulation. When such stimuli are present and cognition is directed towards them, the DMN is suppressed. Changes in this network have been reported in a great deal of neuropsychiatric conditions, most notably in Alzheimer’s disease, depression, bipolar disorder, ADHD, among others [53]. Although it has no direct correlation with the visual process per se, multiple studies in our selection found intranetwork functional connectivity changes within the DMN, as well as internetwork aberrations with the VN.

Frezotti et al., 2016 [24], reported increased FC in parts of the DMN, while Li et al., 2014 [33], reported reduced ALFF. Functional connectivity between the visual cortex and components of the DMN also seems to be lower—Dai et al., 2013 [22], Wang, B et al., 2021 [30]. Wang, J et al., 2016b [26], had the same results for the connectivity between V1 and the DMN connectivity; however, between V2 and the DMN, the results were the opposite. The literature is uncertain about the cause of these changes; however, we can suggest that it may result from failed functional reorganization following clinical worsening of glaucoma and aberrations in the early visual cortices.

### 4.5. RS Changes Correlated with Clinical Measures

Most of the included studies also correlated their main results with various clinical biomarkers for glaucoma stage and severity. The most common such markers were Mean Deviation for visual field restriction, intra-ocular pressure, retinal nerve fiber layer thickness, cup-to-disc ratio, and clinical severity on the Hodapp–Anderson–Parish scale [54]. In summary, reduced functional parameters in the striate and extrastriate visual cortex usually correlated with worse clinical markers. The opposite was mostly true regarding regions and networks outside of the early visual cortices and the visual network, where increased functional parameters correlated with worse biomarkers.

### 4.6. Lateralization of Changes

The reviewed literature seems to suggest that POAG-related functional brain changes in multiple regions are not symmetrical. This asymmetry appears consistently in regions such as ITG, MiFG, and SFG with a prevalence for changes on the right, in IPL and preCG on the left, and in Calc bilaterally. In postCG, Fusiform, and LG, however, there is some divergence regarding laterality. We can propose no meaningful explanation for this discrepancy besides it arising as an artifact due to the direct comparison of different imaging methodologies, which carry different information. We also believe it worthwhile to point out that all papers controlled for handedness and that no studies reported a statistically significant prevalence of glaucoma on one side as compared to the other in the patient population.

### 4.7. Limitations of the Studies Included

Physiological aging as well as the associated neurological, psychiatric, and systemic conditions all inadvertently lead to changes in the functional organization and parameters of the human brain. As POAG is a chronic disease, which is most predominant in (but not limited to) individuals in the latter half of their lives, this can cause a degree of confound when conducting neuroimaging studies. With this in mind, we have only included papers, in which the POAG subject population was controlled against age-matched healthy participants. It is generally accepted that functional networks undergo gradual and progressive changes throughout one’s lifetime; however, there is some evidence that this process significantly accelerates around the age of 60—especially when it comes to the DMN [55]. Among our included papers, 22 had a mean POAG subject age of under 60 (mean age between 35 and 50 in 19 and 50 and 59 in 3), and only 1 had a mean above 60 (71 years in DeMaria et al., 2021 [44]). Furthermore, virtually all studies actively sought and excluded any subject with diagnosed neurological, psychiatric, or non-POAG ophthalmic disorders, with the majority also extending the criteria to systemic diseases like arterial hypertension and diabetes mellitus, as well to the usage of some medication classes. Nevertheless, demographic heterogeneity between studies, mostly stemming from the non-standardized subject inclusion/exclusion criteria, might have an effect on the results.

## 5. Conclusions

In this systematic review, we conglomerated data from 24 papers, examining the resting-state functional changes in patients suffering from POAG. Five broad groups of RS methodologies, resulting in different functional parameters, were used by these papers, namely variations in functional connectivity, arterial spin labeling and cerebral blood flow, regional homogeneity, amplitude of low-frequency fluctuations, and graph network analysis. Based on their results, the following conclusions can be made: POAG leads to reduced functional parameters in the early visual cortices; POAG has effects on higher segments of the visual network; components of the ventral stream are seemingly more often affected as compared to those of the dorsal; a multitude of extravisual networks are also functionally aberrant in POAG patients; and many of these cerebral changes may be correlated clinically with an array of ophthalmological measurements.

As is characteristic for most reviews, however, ours suffers from certain limitations. Most of the papers we have included have differing inclusion and exclusion criteria for patients and controls—different age ranges, glaucoma severity and duration, comorbidities, medication regimen, etc. Furthermore, some methodological heterogeneity within our methodological groups is also present. We should also note the limited scope of this review, as we have not included any task-based data nor such from any other type of glaucoma besides POAG. Further efforts in the field of functional neuroimaging as a whole are required in order to achieve standardization and harmonization of the study protocols and to thereby reduce heterogeneity on a methodological level. Another critical challenge before the field is introduction of normative criteria based on large-scale connectome atlases, which may underpin more successful efforts for clinical translation of the results into clinical reasoning in medical diagnostics.

We believe that this review will contribute to an improved understanding of the neural aspects of glaucoma as well as to the development of a more holistic eye–brain model in this disease. This in turn may be an important cornerstone for the potential emergence of more advanced neuroprotective or neuroregenerative therapeutic approaches, which may improve clinical outcomes.

## Figures and Tables

**Figure 1 biomedicines-13-01402-f001:**
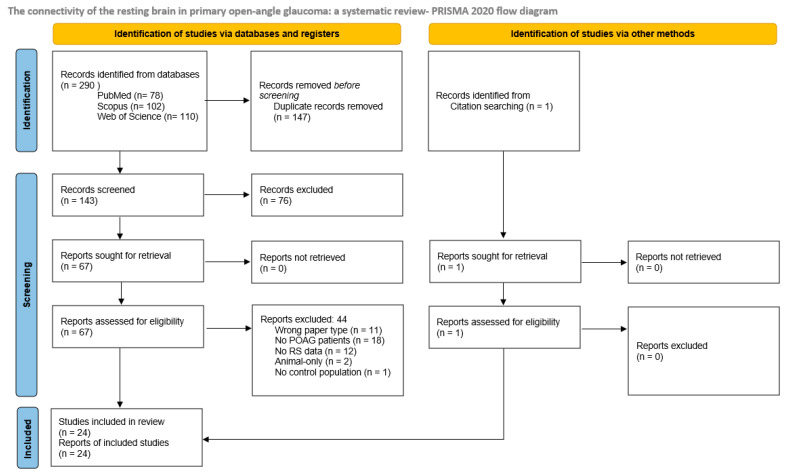
PRISMA flow chart.

## Data Availability

Data on reviewed studies is available upon request.

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
