# Peer review of "The Connectivity of the Resting Brain in Primary Open-Angle Glaucoma: A Systematic Review"

_biomedicines, 2025, doi:10.3390/biomedicines13061402_

Round 1
Reviewer 1 Report
Comments and Suggestions for Authors
In this systematic review, Velkov et all. focused specifically on RS-fMRI analyses performed on the most common type of glaucoma - the POAG, by exploring and comparing their methodologies and results. As molecoular biologist in the field of the molecoular ophtalmology, I appreciated the huge amout of manuscript reviewed. Personally I find the style of this manuscript is to be overly schematic with a long list of studies reported but lacking of a real interconnection between al the studies listed.
Author Response
Response: Thank you for your comment. We agree that the reports reviewed are numerous and therefore the list of studies and illustrative tables are long but we believe a systematic review needs to be thorough as required by the PRISMA guidelines that we have followed as strictly as possible. We have grouped the studies according to their methodology starting with the most abundant group of classical functional connectivity studies based on correlations of BOLD timeseries, followed by the ALFF, ReHo and ASL based reports and finally we have focused on research using advanced graph theoretical analysis, giving insight in the higher order of brain connectivity. We believe that our approach allows us to critically analyze the results of the individual papers in light of the development of various neuroscientific acquisition and analysis techniques, taking into account different sample characteristics (sample size, age and sex distributions, clinical features etc.) as well. This allows us to draw well grounded conclusions about which findings are well supported and which are more controversial.
Reviewer 2 Report
Comments and Suggestions for Authors
This manuscript presents a focused and timely systematic review on resting-state functional MRI (RS-fMRI) alterations in primary open-angle glaucoma (POAG). Given the growing interest in neuroimaging correlates of ophthalmic disease, this work fills an important gap by compiling 24 studies across multiple RS-fMRI methodologies. However, several concerns need to be addressed prior to publication.
- Several included studies report laterality-specific findings (e.g., L ITG, R fusiform), but the draft does not explore potential implications of these asymmetries. It would strengthen the discussion to briefly consider whether lateralized changes are consistent, meaningful, or potentially methodological artifacts.
- Since POAG primarily affects older adults and RS-fMRI signals are age-sensitive, the review should more explicitly discuss whether and how the included studies controlled for age-related confounders (such as cognitive decline, vascular aging). This is critical when interpreting extravisual network changes like those in the DMN.
- While the manuscript clearly states that the literature search was conducted up to Jun. 12, 2024, it does not specify the starting year of the search. Please clarify the full date range to ensure transparency and reproducibility, in accordance with PRISMA guidelines.
- Ensure consistent use of abbreviations (e.g., VN for visual network in Line 198). Define all abbreviations at first use and maintain consistent usage thereafter.
- You mention supplementary materials about your search strategy in section 2.1. Please ensure these are provided.
Author Response
Summary: Thank you for your positive feedback and valuable comments. We shall adress each of them individually below:
1."Several included studies report laterality-specific findings (e.g., L ITG, R fusiform), but the draft does not explore potential implications of these asymmetries. It would strengthen the discussion to briefly consider whether lateralized changes are consistent, meaningful, or potentially methodological artifacts."
Response: Thank you for this suggestion! We’ve added a section to the discussion where we address the laterality issue:
"4.6. Lateralisation of changes
The reviewed literature seems to suggest that POAG-related functional brain changes in multiple regions are not symmetrical. This asymmetry appears consistently in regions such as ITG, MiFG, SFG with a prevalence for changes on the right, in IPL and preCG on the left, and in Calc bilaterally. In postCG, fusiform and LG, however there is some divergence regarding laterality. We can propose no meaningful explanation for this discrepancy besides it arising as an artifact due to the direct comparison of different imaging methodologies, which carry different information. We also believe it worthwhile pointing out that all papers controlled for handedness and that no studies reported a statistically significant prevalence of glaucoma on one side as compared to the other in the patient population."
2. "Since POAG primarily affects older adults and RS-fMRI signals are age-sensitive, the review should more explicitly discuss whether and how the included studies controlled for age-related confounders (such as cognitive decline, vascular aging). This is critical when interpreting extravisual network changes like those in the DMN."
Response: Thank you for pointing to this aspect. This is definitely an important point to be discussed. We have added another new section in the discussion in which we talk about the methods the authors used to control for age (age-matched controls) and for other present morbidities (various exclusion criteria). We also briefly explore the ranges of patient mean age throughout the included articles and mention how their variability may be a study limitation.
“4.7. Limitations of the studies included
Physiological aging, as well as the associated neurological, psychiatric and systemic conditions all inadvertently lead to changes in the functional organization and parameters of the human brain. As POAG is a chronic disease, which is most predominant in (but not limited to) individuals in their latter half of life, this can cause a degree of confound when conducting neuroimaging studies. With this in mind, we’ve only included papers, in which the POAG subject population was controlled against age-matched healthy participants. It is generally accepted that functional networks undergo gradual and progressive changes throughout one’s lifetime; however, there is some evidence that this process significantly accelerates around the age of 60- especially when it comes to the DMN (10.1016/j.neubiorev.2013.01.017). Among our included papers, 22 had a mean POAG subject age of under 60 (mean age between 35-50 in 19 and 50-59 in 3), and only 1 had a mean above 60 (71 years in DeMaria et al., 2021) Furthermore, virtually all studies actively sought and excluded any subject with diagnosed neurological, psychiatric or non-POAG ophthalmic disorders, with the majority also extending the criteria to systemic diseases like arterial hypertension and diabetes mellitus, as well to the usage of some medication classes. Nevertheless, demographic heterogeneity between studies, mostly stemming from the non-standardized subject inclusion/exclusion criteria, might have an effect on the results."
3. "While the manuscript clearly states that the literature search was conducted up to Jun. 12, 2024, it does not specify the starting year of the search. Please clarify the full date range to ensure transparency and reproducibility, in accordance with PRISMA guidelines."
Response: Thank you for addressing this issue. The last search was conducted on the 6th of December 2024, and the search was not limited to any timeframe retroactively. We are not aware of any such requirements in the PRISMA guidelines. Nevertheless, our oldest included study is from September 2013. We found no relevant RS-fMRI studies on POAG from before this date.
4. "Ensure consistent use of abbreviations (e.g., VN for visual network in Line 198). Define all abbreviations at first use and maintain consistent usage thereafter."
Response: Thank you for noticing this inconsistency in our draft. We have carefully checked and corrected the spotted issues.
5. "You mention supplementary materials about your search strategy in section 2.1. Please ensure these are provided."
Response: We apologize for this mismatch. Due to technical issues, the supplementary file has not been properly uploaded to the submission system.